# The effect of housing environment on bone healing in a critical radius defect in New Zealand White rabbits

Patricia Hedenqvist[1]*, Amela Trbakovic[2], Torbjörn Mellgren[3], Caroline Öhman-Mägi[3], Petra Hammarström Johansson[4], Elin Manell[1], Stina Ekman[5], Cecilia Ley[5], Marianne Jensen-Waern[1], Andreas Thor[2]

1 Department of Clinical Sciences, Swedish University of Agricultural Sciences, Uppsala, Sweden, 2 Department of Surgical Sciences, Plastic & Oral and Maxillofacial Surgery, Uppsala University, Uppsala, Sweden, 3 Department of Engineering Sciences, Uppsala University, Uppsala, Sweden, 4 Department of Prosthodontics / Dental Materials Science, The Sahlgrenska Academy, Institute of Odontology, University of Gothenburg, Gothenburg, Sweden, 5 Department of Biomedical Sciences and Veterinary Public Health, Swedish University of Agricultural Sciences, Uppsala, Sweden

* Patricia.Hedenqvist@slu.se

## Abstract

In animal studies on bone healing, the effect of housing space and physical activity are seldom taken into account. Bone formation was evaluated in New Zealand White rabbits (mean ± SEM BW: 3.9 ± 0.11 kg) with a critical bone defect after 12 weeks of rehabilitation in pair-housing in 3 m² large floor pens (Floor, n = 10) or standard single housing in 0.43 m² cages (Cage, n = 10). In the randomised full-factorial study, a bone replica of calcium phosphate cement (CPC, n = 10) or autologous bone (AB, n = 10) was implanted in the unilateral 20 mm radius defect. Post-mortem, the oxidative capacity was measured by citrate synthase (CS) activity in *M. quadriceps* and the defect filling volume and density evaluated by microcomputer tomography (µ-CT). Histology sections were evaluated by subjective scoring and histomorphometry. Fourteen rabbits remained until the end of the study. Group Floor (n = 7; 3 CPC + 4 AB) had a higher CS activity and a larger bone defect filling volume and lower density by µ-CT measurements than group Cage (n = 7; 3 CPC + 4 AB). Three out of four rabbits in AB-Floor presented fusion of the defect with reorganisation of trabecular bone, whereas three of four in AB-Cage showed areas of incomplete healing. Floor rabbits had a higher score of bony fusion between the radius and ulna than Cage rabbits. There were no differences between groups in histomorphometry. The study found that a larger housing space increased physical activity and promoted bone formation.

## Introduction

The rabbit is one of the most commonly used animal species in orthopaedic research, due to its small size, low cost and translational value [1]. The rabbit serves as an intermediate species in studies of bone healing, before large animal implantation. The rabbit's long bones are large

**Data Availability Statement:** All relevant data are within the manuscript and its Supporting Information file.

**Funding:** PH+MJW: Swedish Research Council: K2012-79X22022-01-3. The funders had no role in study design, data collection and analysis, decision to publish, or preparation of the manuscript.

**Competing interests:** The authors have declared that no competing interests exist.

enough for evaluation of material handling properties in comparison to smaller rodents [2]. In orthopaedic, as well as other types of studies, rabbits are often housed individually in cages, despite recommendations of social housing and larger spaces [3]. The minimum floor requirement in the European Directive 2010/63/EU for single housing of rabbits weighing 4–5 kg is 0.42 $m^2$, with a minimum cage height of 45 cm. In studies of bone regeneration, the recommendation is to use New Zeeland White (NZW) rabbits > 6 months of age, to ensure that growth plates of the long bones are already closed [4]. The rehabilitation times in bone defect studies can last up to 52 weeks [5–7], during which most rabbits are kept singly housed in cages. This is not only a welfare issue but may also have an impeding effect on bone quality and formation, and lead to underestimations of the effect of the experimental treatment. Rehabilitation under more physiologic conditions, which increase the mechanical load on the bones through exercise, could improve bone formation, and thus the validity of the animal model. The effect of housing conditions on bone formation and healing in rabbits has not been studied. There are however studies showing an increase in cortical bone thickness but not in bone strength in non-injured rabbits housed singly in larger cages or in pairs [8, 9]. It has further been found that complete immobilization of a limb in rabbits causes bone loss in as little as 6 weeks [10]. A study in rats showed that intense exercise increased the amount of new bone and the bone mineral density in a tibia defect [11].

Our hypothesis was that a 12-week rehabilitation period after grafting of a unilateral critical radius defect in adult rabbits would lead to increased skeletal muscle oxidative capacity and improved bone formation if rabbits were pair-housed in floor pens of 3 $m^2$, compared with single housing in cages of 0.43 $m^2$. Floor housing was expected to increase bone volume and density in the defect area and in the contralateral uninjured bone. The 20 mm radius defect was grafted with a three dimensional (3D) modelled calcium phosphate (CPC) block, or particulated autologous bone (AB).

## Materials and methods

The study was granted permission by the Uppsala Ethics Committee for Animal Experiments (C131/11).

### Animals

Twenty mature female New Zealand White rabbits (Lidköpings kaninfarm, Lidköping, Sweden), 9–10 months of age and with a mean (± SEM) body weight of 3.9 ± 0.11 kg were purchased from a breeder. The breeder colony was free from specified pathogens by health monitoring according to European recommendations [12]. The rabbits were housed singly or in pairs in wire floor cages with an area of 0.35 $cm^2$ at the breeder establishment.

Upon arrival at the university animal facility, the rabbits were acclimatized for three weeks. During the third week, they were accustomed to the presence of the research staff 5 min per rabbit and day, and to handling by daily recording of body weight.

### Study design

A 2 x 2 factorial design was used, with Housing (Floor or Cage) and Material (CPC or AB) as factors (Table 1). The rabbits were randomly assigned to the groups (n = 5 per group).

### Housing

Rabbits were either pair housed in floor pens (Floor, n = 10), or singly housed in cages (Cage, n = 10). The floor pens measured 3 $m^2$ and were made of steel mesh partitions (Troax®

**Table 1. Study design.**

| Material | Calcium phosphate (CPC) | Autologous bone (AB) |
| --- | --- | --- |
| Housing | | |
| Floor pen (in pairs) | n = 5 | n = 5 |
| Cage (single) | n = 5 | n = 5 |

Classic, Office Troax AB, Hillerstorp, Sweden) and a plastic floor (Fig 1). The floor pens were equipped with aspen bedding (Tapivei, Finland) and plastic houses for hiding (Biltema, Sweden). The cages (EC2, Scanbur, Karlslunde, Denmark) had a perforated plastic floor area of 0.43 m², a height of 65 cm and a combined shelter and elevated resting area (Fig 2). Floor pens and cages were cleaned once a week. All rabbits had access to autoclaved straw and hay and tap water *ad libitum*. Rabbits were fed 160 g of commercial pelleted rabbit feed (Lactamin K1, Lantmännen, Sweden) per day. The light cycle was 12:12, room temperature 18.5–20.7˚C and the relative humidity 42–66%.

## Graft material

The CPC implant was manufactured with a 3D printing method and designed as a replica of a rabbit radius based on a μCT scan (Trbakovic/Mellgren *et al*, in manuscript). The radius replica was 3D-printed (Makerbot 5th generation, Makerbot Industries LLC, Brooklyn, New York, USA) using a polylactic acid filament (True White PLA, Makerbot). A casting mould using the radius replica was then produced using silicon rubber (Elastosil M4601 A/B, Wacker

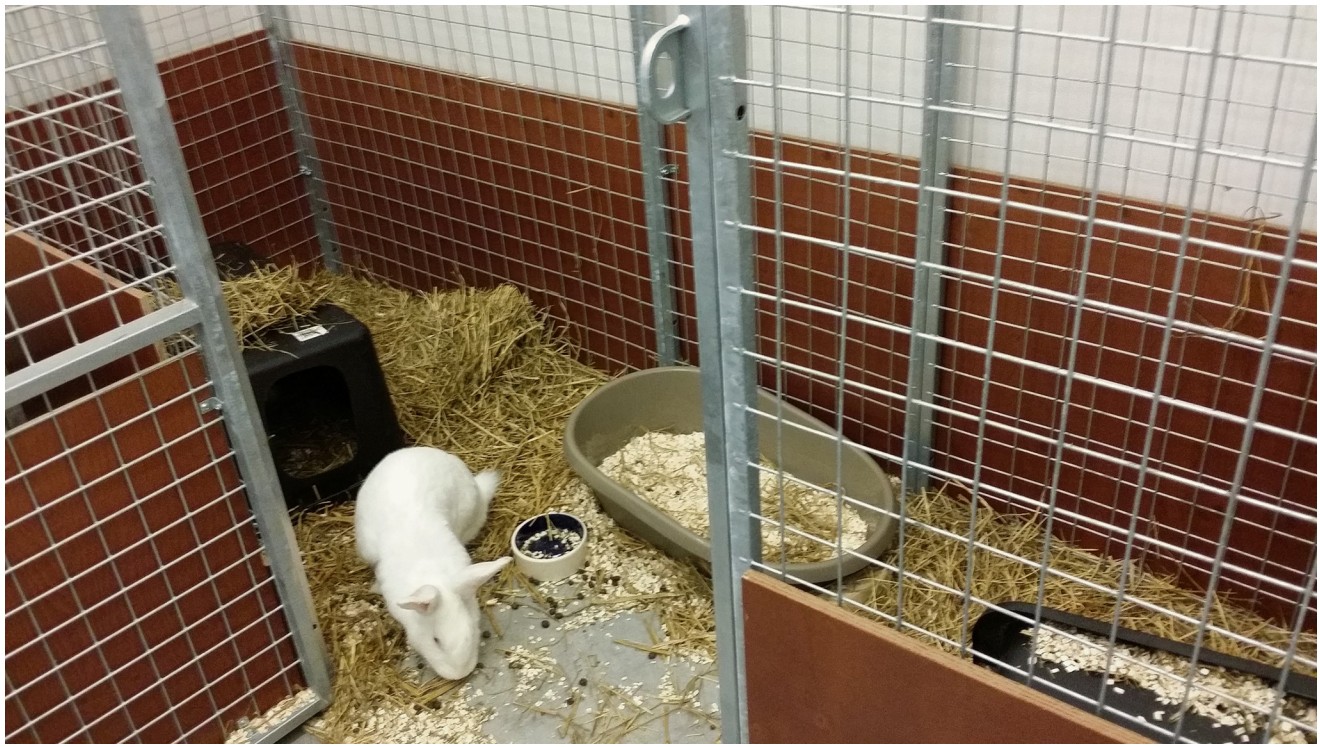

**Fig 1. Pair housing of rabbits in floor pens of 3 m².** Pens were provided with aspen bedding, autoclaved straw and hay, two plastic houses and a plastic tray with bedding. Pens were cleaned weekly.

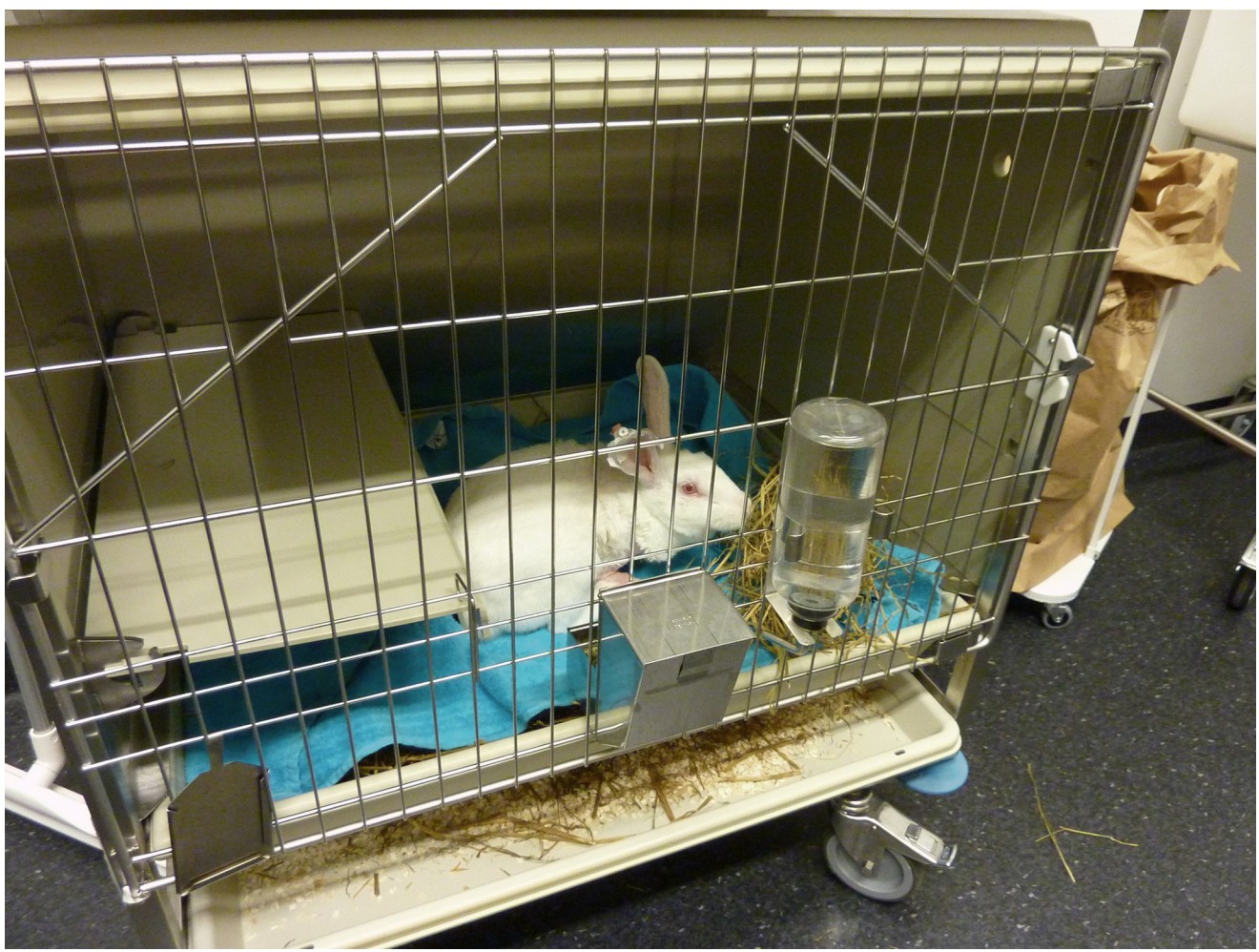

**Fig 2. Single housing of rabbits in cages of 0.43 m2.** Cages were provided with autoclaved straw and hay and a shelf to hide under or sit on. Cages were cleaned weekly. A towel was provided in the cage until the rabbits were fully recovered from surgery.

Chemie AG, Munich, Germany). The model of the medullary cavity was 3D-printed (Makerbot Replicator 2) using a water-soluble polyvinyl alcohol filament (PVA water soluble, Makerbot). The CPC was prepared by mixing β-tricalcium phosphate (β-TCP, Sigma-Aldrich) with monocalcium phosphate monohydrate (MCPM, Scharlau) in the ratio of 55.2:44.8, the powder mixture was then mixed with glycerol with a powder to liquid ratio of 3.9 g/ml. After preparation, the cement was immediately injected into to the radius moulds and the model of the medullary cavity was pushed into the centre of the cement filled moulds. The implant was left to cure submersed in sterile and distilled water at 37 ˚C for 4 days, the water was however replaced after 2 days. The CPC implants were carefully removed from the moulds and checked for any visible defects and left to dry in room temperature for 24 hours, before they were packaged and sterilized in an autoclave at 121 ˚C for 20 minutes.

### Anaesthesia and preparation for surgery

The rabbits were clinically examined on the day of surgery. General anaesthesia was induced with medetomidine (0.25 mg/kg, Domitor 1mg/ml, Orion Pharma AB Animal Health,

Danderyd, Sweden) and ketamine (15 mg/kg, Ketalar 50 mg/ml, Pfizer AB, Sollentuna, Sweden), mixed and administered subcutaneously (SC). After induction of anaesthesia, carprofen (5 mg/kg, Norocarp, N-vet AB, Uppsala, Sweden) and metoclopramide (0.5 mg/kg, Primperan, Sanofi AB, Stockholm, Sweden) were administered SC and ceftiofur (10 mg/kg, Excenel vet, Orion Pharma, Sweden) intramuscularly (IM). One 18G catheter (Venflon, B&D AB, Stockholm, Sweden) was placed in an ear artery for measurement of blood pressure and blood sampling, and one 20G catheter in an ear vein, for the administration of Ringer Acetate (10 ml/kg/h) during surgery. The catheters were filled with heparin lock solution (100 IE/ml, LEO Pharma AB, Malmö, Sweden) when not in use. Ropivacain (10mg/ml, Fresenius Kabi AB, Uppsala, Sweden) was injected SC at a dose of 10 mg/kg in the axillary brachial nerve plexus area and the surgical area was aseptically prepared.

During surgery, the rabbits were administered isoflurane as needed (0–3%) and oxygen (1.5 L/min), via a larynx mask (V-gel, large, Docsinnovent, London, UK). The mask was connected to the anaesthesia machine (Anmedic Q-Circle System; Anmedic AB, Stockholm, Sweden) via a paediatric breathing system (Intersurgical Ltd, Wokingham, UK). The rabbits were placed on a heating mat. Rectal temperature and arterial blood pressure were measured continuously (CS/3, Datex-Engstrom, Helsinki, Finland).

## Surgery

An experienced maxillo-facial surgeon (AnT) performed the osteotomies; a 20 mm long piece of the radius was removed mid-shaft with a Pietzo burr (Mectron, Carasco, Italy) under continuous saline cooling. A prefabricated template (4x6x20mm) was used as a guide. The removed bone was replaced with the same size CPC implant (n = 10, Fig 3) or AB (n = 10), after particulation into 1–3 mm bone chips using a manual bone grinder (The R. Quétin Bone-Mill, KLS Martin, Jacksonville, FL, USA). A collagenous membrane (25 x 25 mm, Geistlich Bio-Gide®, Geistlich Pharma AG, Wolhusen, Switzerland) was applied to form a cylinder around the AB in the defect area (Fig 4). No other fixation was used. The soft tissue was closed in separate layers using resorbable Monocryl 5–0 sutures (Ethicon, Johnsons & Johnson AB, Solna, Sweden). The surgery lasted 60–90 min.

## Postoperative care

Following surgery, the rabbits were placed in a heated chamber (26˚C) until fully awake. Every 15 min, the general condition, respiration and colour of the mucous membranes were examined until the rabbits were fully recovered. If the righting reflex had not returned within 15 min, atipamezol (5mg/ml, Antisedan, Orion Pharma AB Animal Health) was administered SC at a dose of 0.25 mg/kg. After the local nerve block had subsided, (90–180 min postoperatively), buprenorphine (0.03 mg/ml, Vetergesic, Orion Pharma AB Animal Health) was administered SC (0.05 mg/kg) or by the orotransmucosal route (OTM, 1.5 mg/kg), and repeated after 8 h. Blood samples were collected from the arterial catheter before and during 9 h following the first bupenorphine administration, for measurement of buprenorphine plasma levels [13]. Thereafter, buprenorphine was administered SC twice daily in a tapered dose (0.05–0.03 mg/kg) and carprofen SC once daily (5 mg/kg), for the three first postoperative days. Metoclopramide (5mg/ml, Metomotyl, Virbac Danmark A/S, Kolding, Denmark) was administered SC 12 and 24 h postoperatively at 0.5 mg/kg. Rabbits that did not start to eat within 12 h postoperatively were syringe fed with a complete powder feed (Critical Care, Oxbow, UK) and administered Ringer Acetate solution SC as needed (10 ml/kg). Rabbits were clinically examined and weighed daily for the first postoperative week, and thereafter weekly until the end of the study.

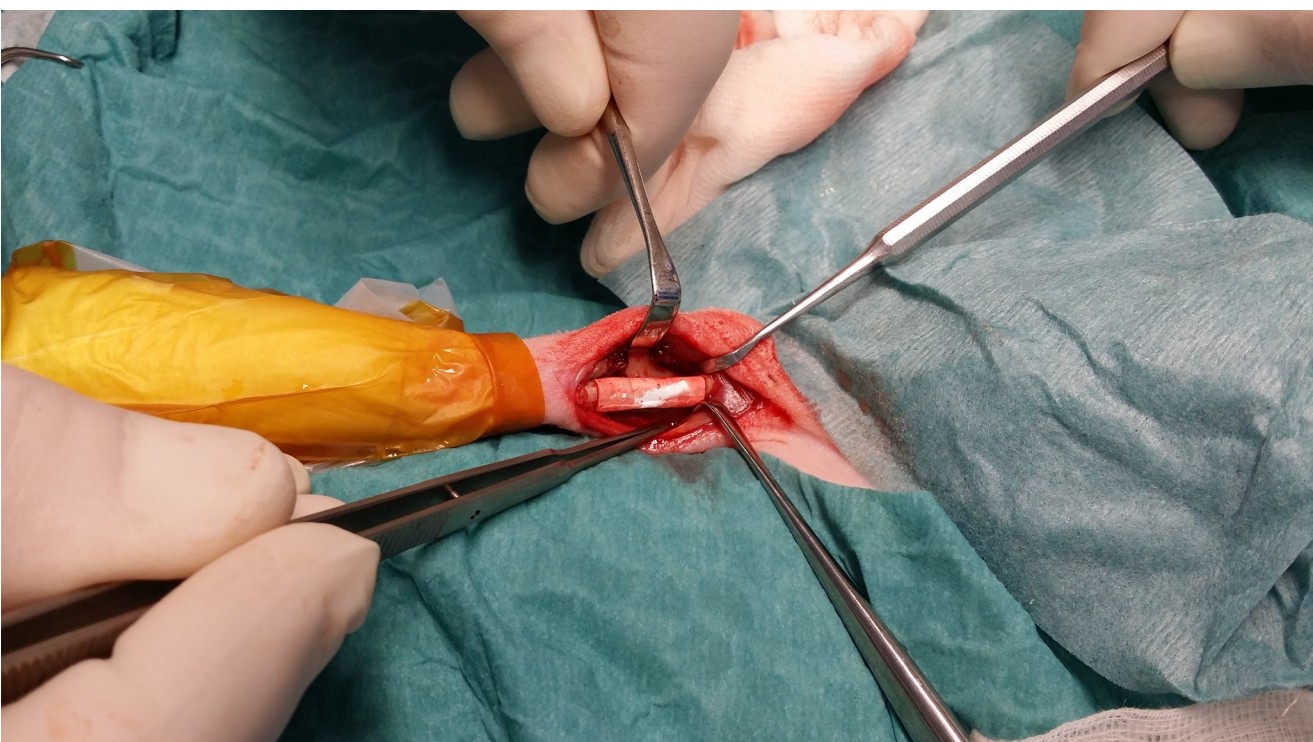

**Fig 3. Picture of a bone replica of calcium phosphate cement (CPC) in place of a 20 mm mid-shaft radius defect in a NZW rabbit.**

## Post-mortem analyses

The rabbits were euthanized with an overdose of pentobarbital (Allfatal vet, Omnidea Stockholm) by injection in the ear vein and both front legs removed for sampling of bone tissue.

## Muscle oxidative capacity

A 0.5 x 0.5 cm piece of muscle tissue was cut out of the *M. quadriceps* immediately after euthanasia, frozen in liquid nitrogen and stored at -80 C˚. The muscle tissue samples were freeze-dried overnight using a Scanvac CoolSafe™ (LaboGene ApS, Allerød, Denmark). The measurement of citrate synthase (CS) activity followed a protocol described by Alp *et al* [14] and modified by Essén *et al* [15]. The muscle was dissected free of blood, fat and connective tissue under a microscope and cut into small pieces. Extraction buffer was mixed with a sample (150 μl/ mg tissue) and homogenized with a Bullet Blender (Next advance Inc. New York, USA) S1 Dataset.

The supernatant was analysed by spectrophotometry (BeckmanDU-650, Grizzly analytical, Sebastopol CA, USA) after mixing with a reagent of DTNB-buffer:acetyl-CoA and addition of oxaloacetate (see S1 Dataset).

## Micro-computer tomography (μ-CT)

Both front limbs were removed and trimmed using a band saw (KT-400, Klaukkala, Finland) and immersed in 10% neutral-buffered formalin fixative. The sample size was 35 x 10 mm.

For the measurement of the defect filling volume (Vol) and mineral density (MD), micro-computed tomography (Micro-CT, Skyscan 1172, Bruker MicroCT, Kontich, Belgium) was

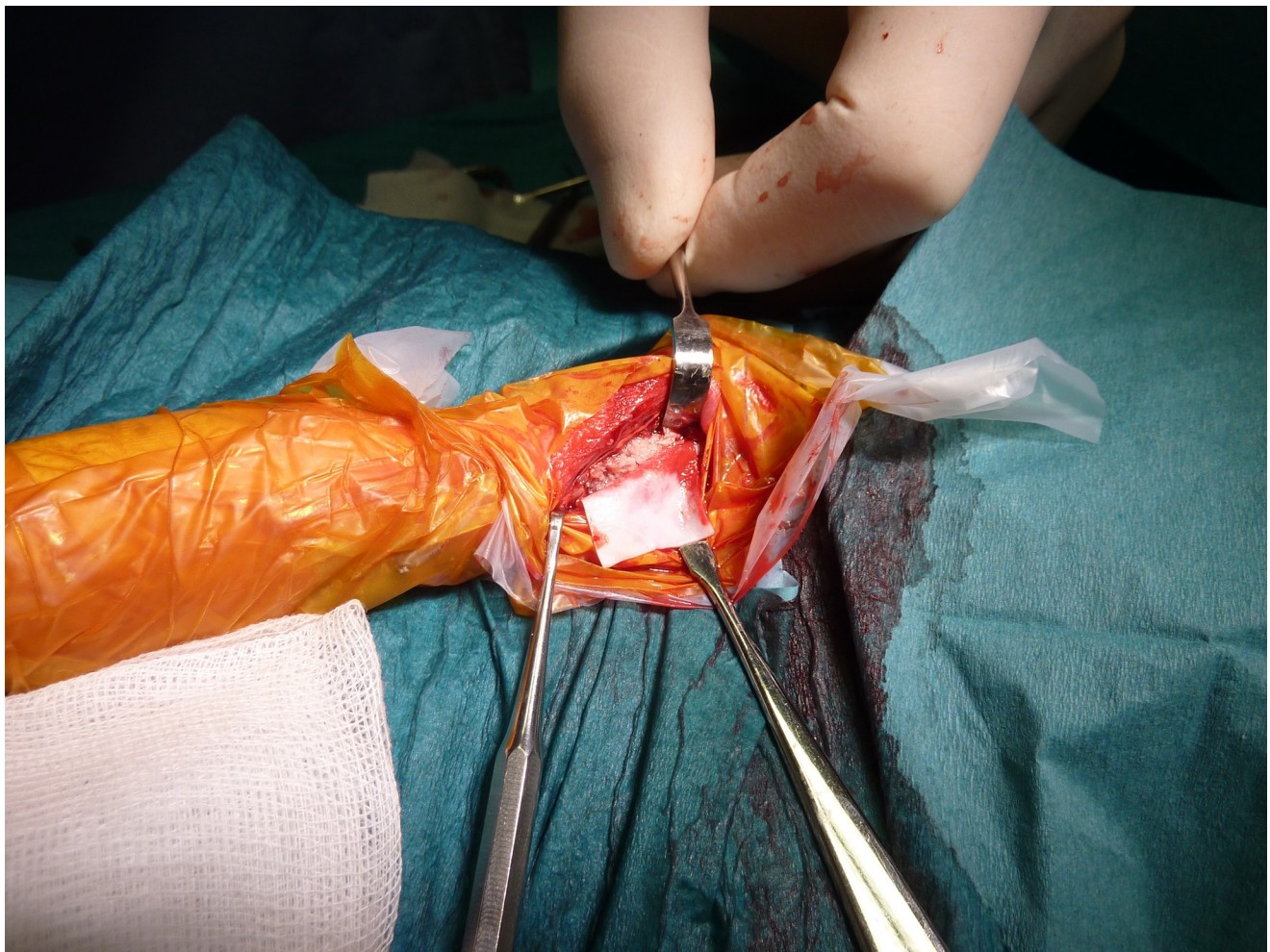

**Fig 4. Picture showing a collagenous membrane to be fitted around bone chips in place of a 20 mm mid-shaft radius defect in a NZW rabbit.**

used. The bone was kept in formalin inside a cylindrical PMMA [poly(methyl methacrylate)] sample holder during scanning. A voltage of 80 kV, a current of 124 µA and a 0.5 mm aluminium filter were used to acquire images with an isotropic pixel size of 9.0 µm$^2$. Two MD reference samples (medium 8–16 mm MD phantom set, Bruker MicroCT, Kontich, Belgium) were scanned once per day to quantify the MD of the defect filling and to achieve inter-study comparable results. Cross-sections were reconstructed using NRecon (Bruker MicroCT, Kontich, Belgium), defect volume and MD measurements were performed with CTAn (Bruker MicroCT, Kontich, Belgium) and 3D models were visualized using CTVox (Bruker MicroCT, Kontich, Belgium). Measurements were performed on the ulnar and radial bone and the remaining graft material, on the operated side. The measurements were performed in the central part of the defect (Op 10), as well as in the entire defect area (Op 20). In both cases, the adjacent ulna and any surrounding new bone was included (Fig 5). Measurements were performed in the equivalent area on the contralateral side (Intact 10 & 20).

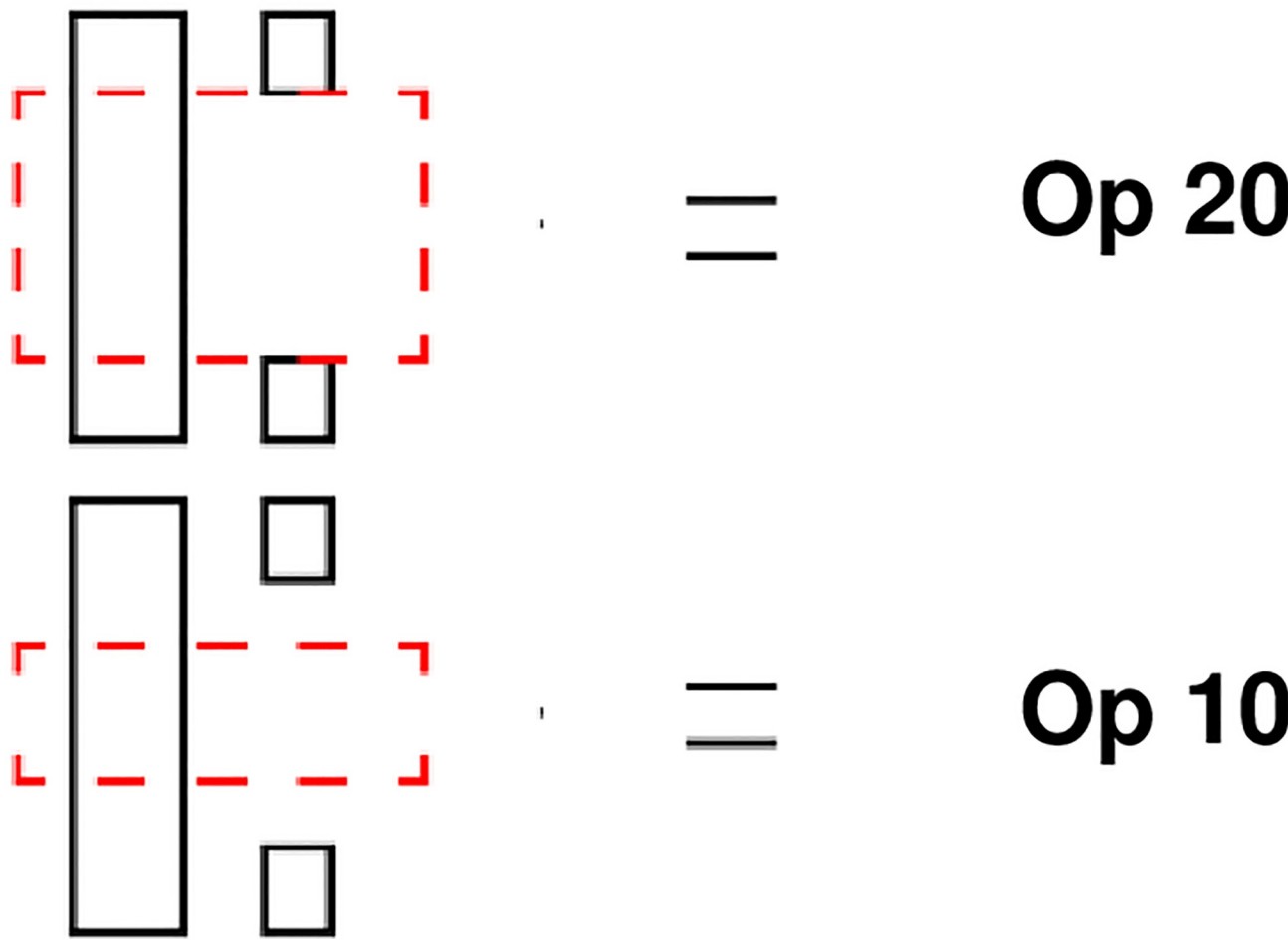

**Fig 5. Measurement of volume and density with μ-CT in a critical radius defect area 12 weeks after treatment with a synthetic material or bone.** Op 20 and 10 show the volume of measurement of the ulna and the radius defect area including any proliferation of bone. 20 mm encompasses the whole defect. 10 mm encompasses the centre of the defect.

### Qualitative histology

Routine and standard preparation of undecalcified cut and ground sections followed the guidelines proposed by [16] and [17]. Radiographs of the bone samples were used to guide the preparation of sections for histology (Fig 6).

Two transversal sections were cut in the centre of the defect. Additionally, two longitudinal sections were cut, one from the centre to the proximal, and one from the centre to the distal end of the defect, extending into the intact border area. Sections were stained with toluidine blue [18] and examined qualitatively with a light microscope (Nikon eclipse E600) by two veterinary pathologists (SE and CL) and one of the authors (AmT), who after individual

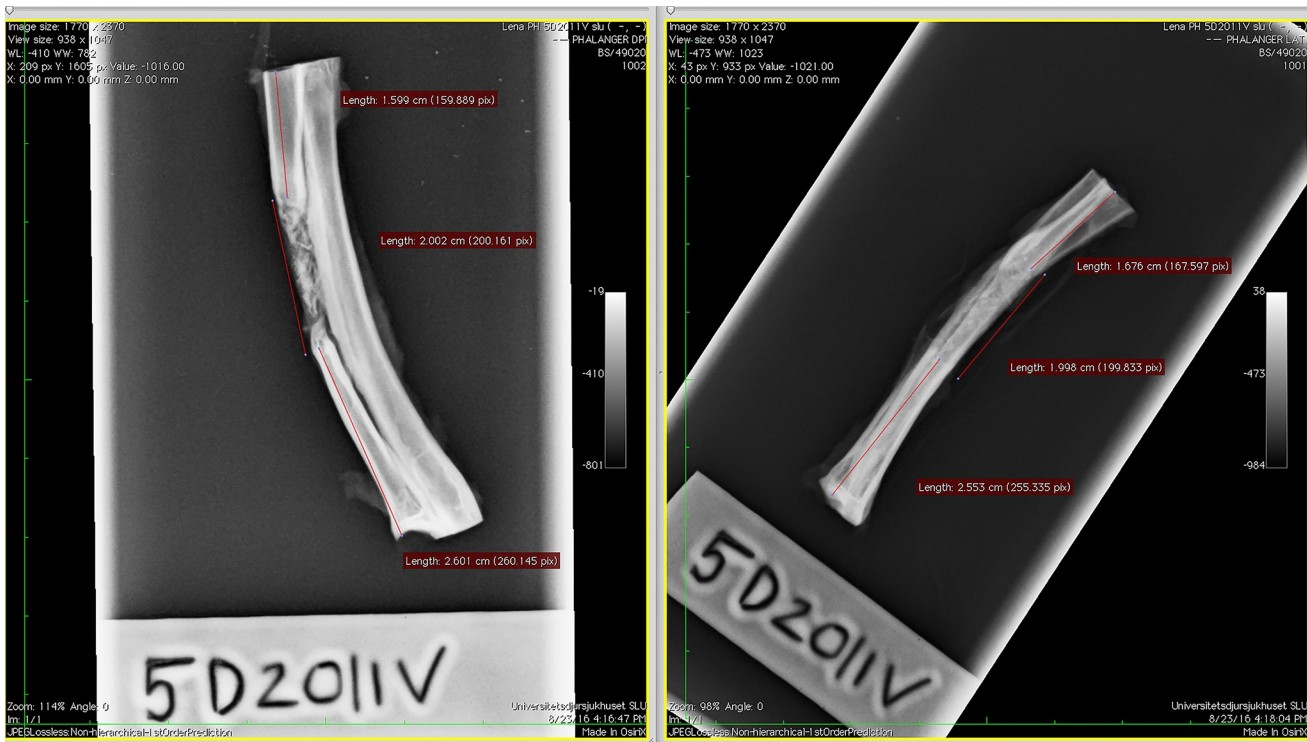

**Fig 6. Radiographs of the radius and ulna sample from one rabbit, harvested 12 weeks after surgery.** The defect and the position of the defect have been measured and marked to guide the preparation of sections for histology.

assessment agreed on a consensus scoring. A scoring system from Bigham-Sadegh and Oryan [1] was used after modification. The longitudinal sections were examined for new bone formation at defect ends (graft integration) (a), cortical integrity (b), appearance of cancellous bone (c), presence of inflammation (d), defect filling (e) and bony fusion with ulna at the defect (f), as well as bony fusion between ulna and radius adjacent to the defect (g) (Table 2). The transversal sections were evaluated for categories b, c, d and f (Table 2). The higher the scores, the

**Table 2. Modified system for subjective scoring of radius defect area by histology [19].**

| Categories | Scoring grade | | | |
|---|---|---|---|---|
| | 1 | 2 | 3 | 4 |
| a) New bone formation associated with defect ends (graft integration) | No integration | One edge integrated | Both edges integrated | - |
| b) Cortical integrity (peripheral surface) | Periosteal fibrosis | Initiation of formation of lamellar bone | Reorganisation in majority | Complete organization |
| c) Cancellous (trabecular) bone | Woven bone associated with graft | Lamellar and woven bone associated with graft | Lamellar and woven bone–no graft seen | Complete reorganisation of trabecular bone |
| d) Inflammation | Multiple areas of > 20 monocytes in a high-power field (x400) | No inflammation | - | - |
| e) Filling of the defect | Filled with graft, bone and fibrous tissue | Areas of fibrocartilage in the bone | Only bone fill the defect | - |
| f) Bony fusion with ulna at the defect | No fusion | Fusion | - | - |
| g) Bony fusion between ulna and radius adjacent to defect | No fusion | Fusion | - | - |

better the healing. A sum of the transversal and longitudinal scores was calculated from each animal.

## Histomorphometry

Longitudinal sections were photographed with objectives of x1 magnification with a Nikon E600 light microscope and a Nikon DXM 1200 camera (Nikon Instruments, Melville, NY, USA) connected to a PC. All slides were photographed under the same conditions for exposure time, light intensity, and camera gain. Analyses were performed by one evaluator (AmT) using NIS-elements imaging software (NIS-elements Basic Research, Nikon, Tokyo, Japan). White balance adjustments, calibrations, and transformations of the photographs into binary images were performed for each captured image. The measurements were performed on an area restricted by the defect, the ulnar bone and the outer border of the graft material (Fig 7). The total area and the area filled by graft material and bone (binary area) were measured. The fraction of binary area:total area was calculated as percentage.

## Statistical evaluation

Data was evaluated with InVivoStat (Bate ST and Clark RA 2014). Continuous data were examined for normality and homogeneity. Body weight data were analyzed using a 3-way Repeated Measures mixed model, with Housing and Material as factors, and Time point as the repeated factor (pre op, 10 d and 12 w post op). Other continuous data were analysed using 2-way ANOVA, with Housing and Treatment as factors. Post-hoc tests were performed using Holm's procedure. Scores (µ-CT, histology) were compared between groups with Mann-Whitney test. For evaluation of histology, scores for each category were compared between housing groups with the Mann-Whitney test, separately for transversal and longitudinal sections. For the transversal categories, a mean was calculated from the two measurements. Additional to the comparison for each category, sums of scores from all categories, either transversal or longitudinal sections, were calculated for each animal, and compared between housing groups.

Continuous data are presented as mean ± SEM, other data as median (min-max). The significance level was set at a P-value of <0.05.

# Results

## Animals

The rabbits were easily frightened, and the behaviour did not change much during the three-week-acclimatisation period. Six rabbits were lost to the study. Rabbit #1 was found dead without prior signs of disease before the study began. Necropsy revealed severe acute haemorrhagic enteritis, typhlitis and hepatic lipidosis. Rabbit #2 died from severe acute lung oedema during anaesthesia, shortly after intubation. Rabbit #3 was euthanized due to seizures during blood sampling after surgery. Rabbits #4–5 were found dead on the day after surgery; one had focal encephalomalacia in the hippocampus area and one signs of haemorrhagic shock. Lastly, rabbit # 6, in group Floor fractured the ulna on the operated side two days after surgery and was euthanized. The rabbit was the heaviest of all (5 kg).

This left 14 rabbits for the remainder of the study, of which seven were in group Floor and seven in group Cage. In each Housing group, three rabbits were in group CPC and four in AB.

## Body weight

Housing had no overall effect on body weight (p = 0.22, F = 1.75) and there were no overall interactions with Treatment (p = 0.075, F = 0.075), nor Time point [(pre op, 10 d and 12 w

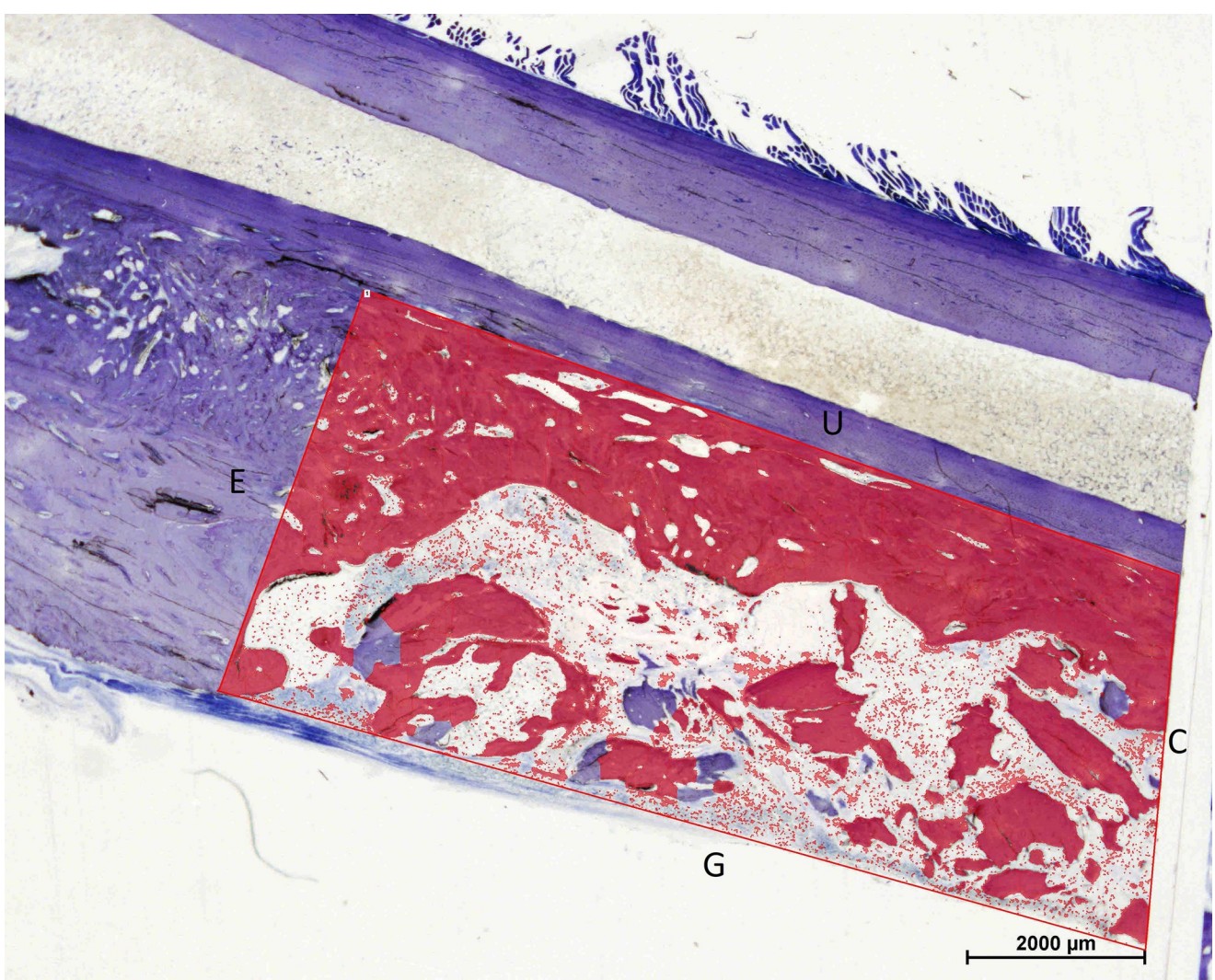

**Fig 7. Outline of the distal part of the area of interest for the histomorphometric measurements.** The total area was outlined by the end (E) and the centre (C) of the defect. Longitudinal borders were the graft border (G) and the ulnar bone (U). The area that is coloured red represents the binary area (area with graft material and bone).

post op) p = 0.46, F = 0.82]. Time point had an overall effect (p<0.001, F = 26.21). The body weight was lower 10 d after surgery compared with pre op and 12 w post op in both group Floor and Cage (Fig 8). The average 10 d body weight loss was 6%.

## Clinical evaluation

All rabbits put weight on the operated leg immediately after surgery. Two weeks post-surgery 11 out of 14 rabbits were limping slightly. Rabbits with AB seemed more swollen in the area of surgery than those with CPC. After 12 weeks, all but two rabbits were walking without a limp and two walked with a slight limp.

## Muscle oxidative capacity

There was no overall effect of Housing on CS activity and no interaction with Treatment (two-way ANOVA, p = 0.057, F = 4.64 and p = 0.47, F = 0.57, respectively). The CS activity was

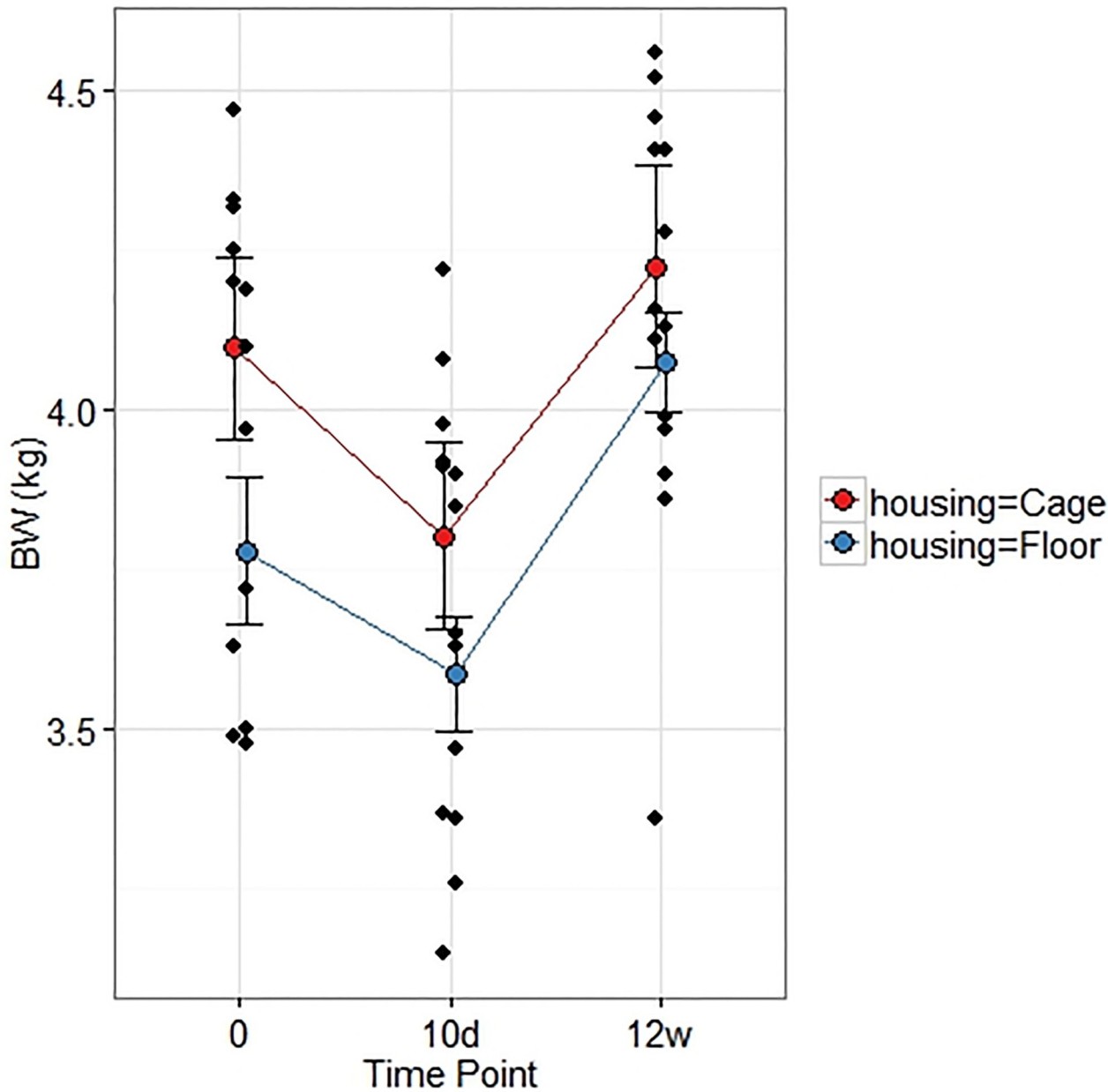

**Fig 8. Mean and individual body weights.** Means (SEM) pre op (0), 10 d and 12 weeks post creation of a critical radius defect in rabbits housed in cages or in floor pens. Mean body weights were lower at 10 d post op compared with pre op in group Floor (p = 0.029) and group Cage (p = 0.003), as well as compared with 12 w post op in group Floor (p = 0.0004) and group Cage (p = 0.0004).

however higher in group Floor than group Cage (p = 0.041, CIs [(52,72), (37, 57)] by one-way ANOVA) Fig 9.

## μ-CT

The CPC implants showed multiple fractures and slight dislocations in both groups Cage and Floor (Fig 10). There were no interactions between Housing and Treatment on Vol (Op 10:

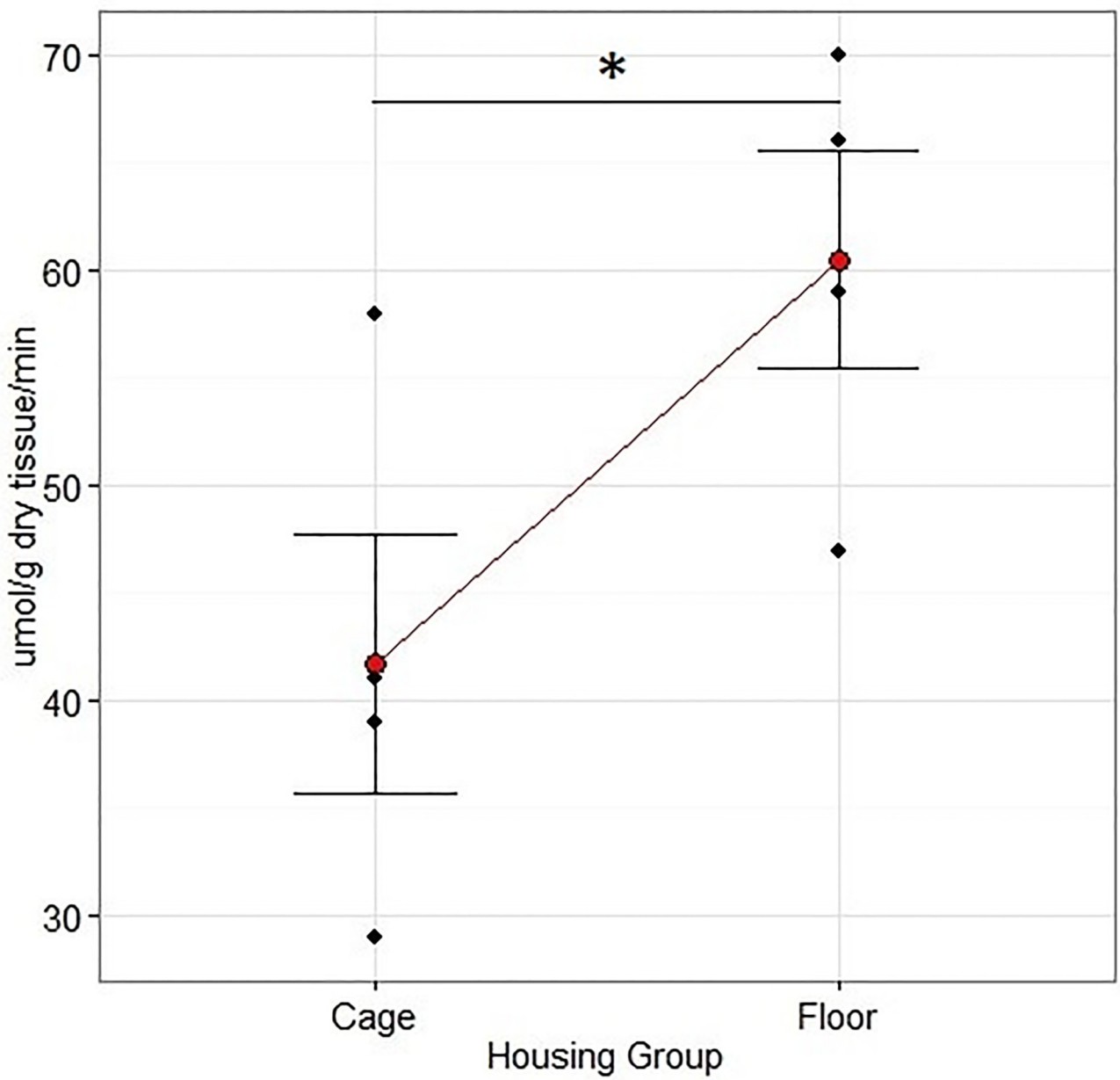

**Fig 9. Rabbit CS activities in the *M. quadriceps* after 12 weeks of rehabilitation following a critical unilateral radius defect.** The animals were housed in floor pens of 3 m$^2$ (n = 7) or cages of 0.43 m$^2$ (n = 7). CS activity was higher in group Floor than in group Cage (p = 0.041, F = 4.64, one-way ANOVA). Data are mean + SEM.

p = 0.25, F = 1.49; Op 20: p = 0.18, F = 2.12) or BMD (Op 10: p = 0.72, F = 0.13; Op 20: p = 0.85, F = 0.04) of the defect filling. Group Floor had a larger Vol and lower MD for Op 20 (Figs 11 and 12), but not Op 10 (Table 3), compared with group Cage. There were no differences in Vol or MD between group Cage and Floor on the contralateral uninjured side for either Intact 10 or Intact 20 (Table 3).

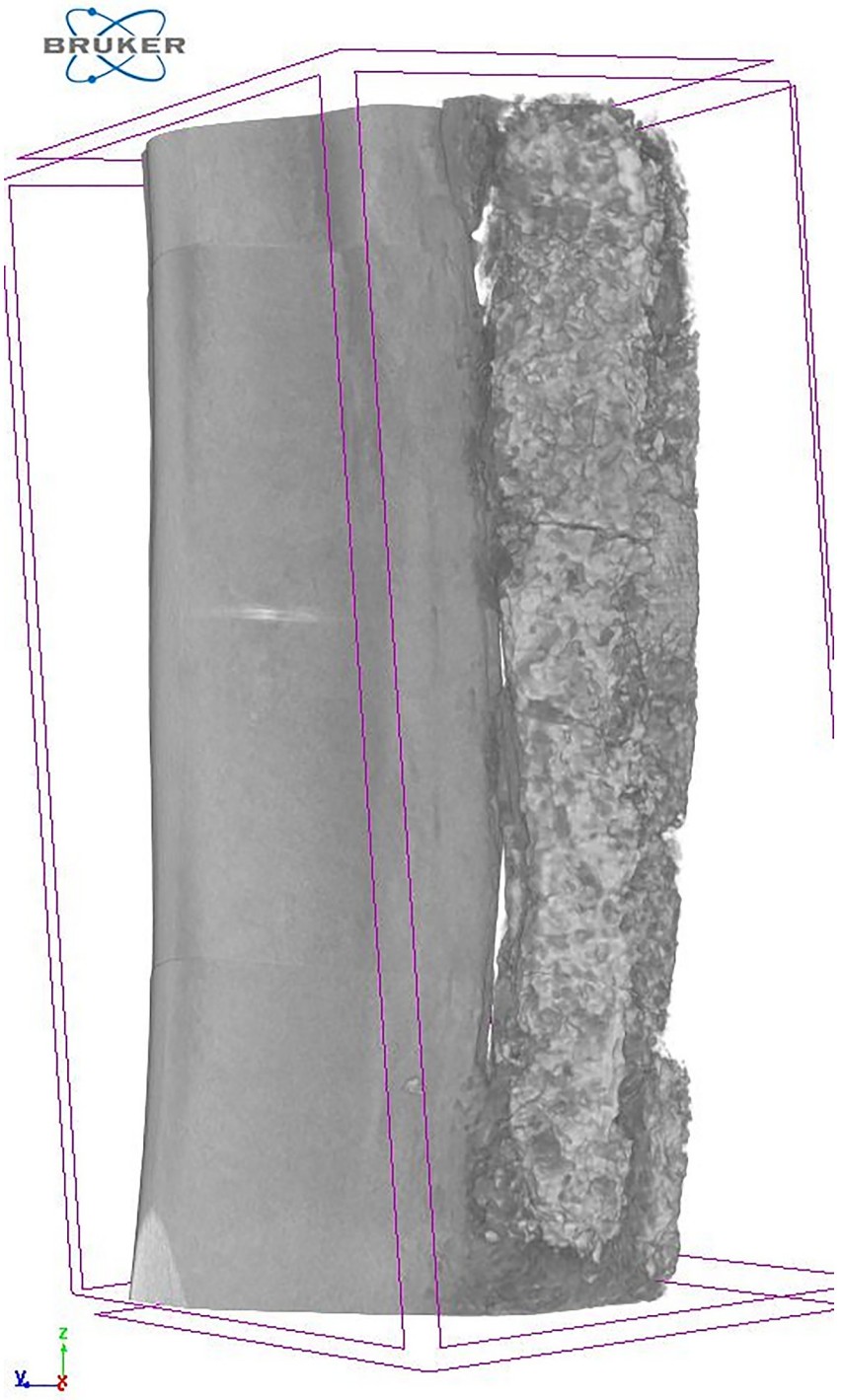

**Fig 10. μ-CT image of the defect 12 weeks after surgery showing CPC implant with visible fractures and slight dislocation.**

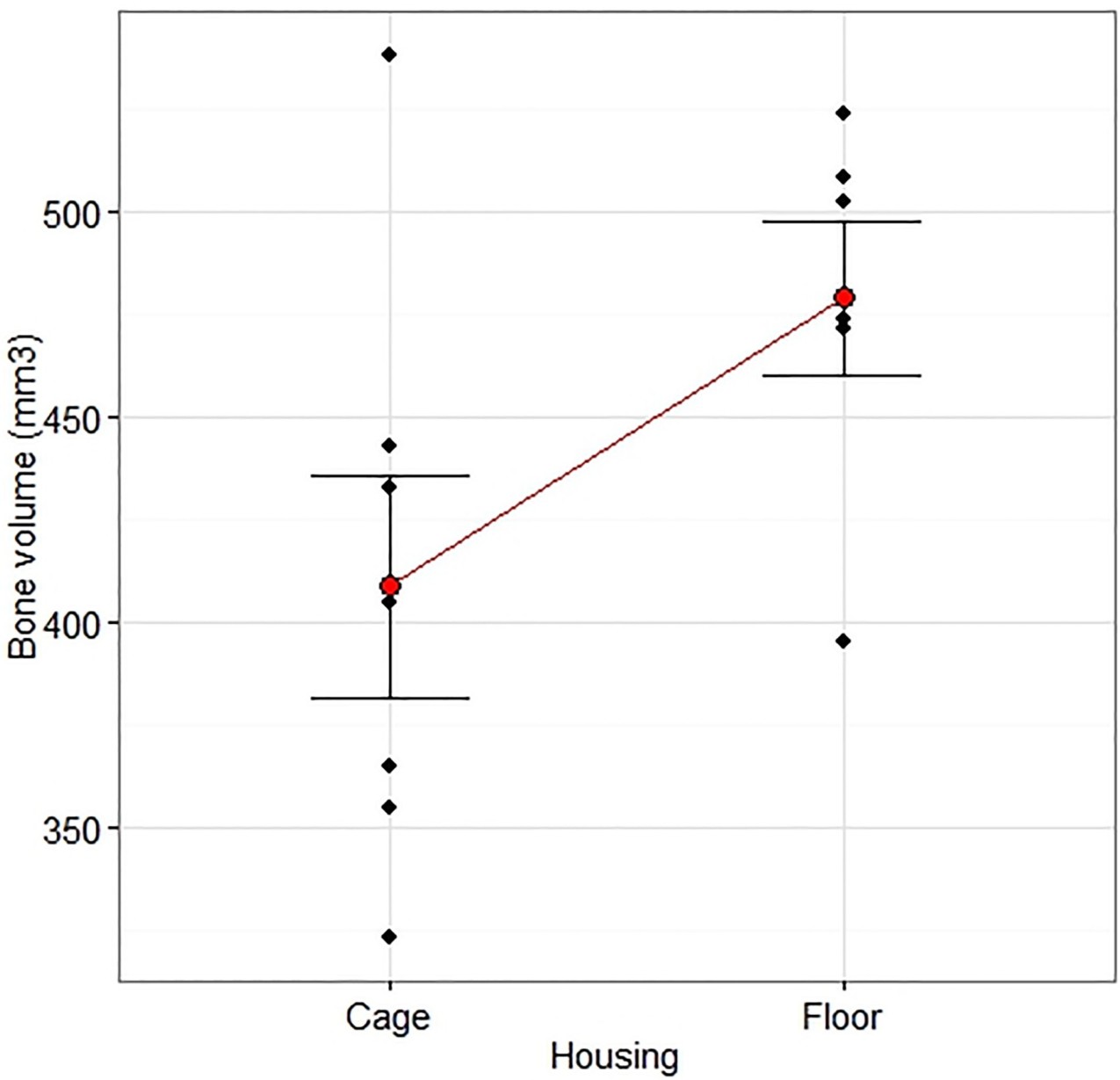

**Fig 11. Volume measured by μ-CT.** The effect of Housing on the defect volume measured by micro-CT 12 weeks after creation of a critical radius defect treated with a calcium phosphate compound or augmented bone. The measurement was performed along the entire defect length (20 mm) and included the ulnar bone. There was an overall effect of Housing (p = 0.028, F = 6.59) that did not interact with Treatment (p = 0.18, F = 2.12). The volume was larger in group Floor (n = 7) than group Cage (n = 7), 95% CIs [355, 454], [436, 535], p = 0.028. Two-way ANOVA with Housing and Material as treatment factors followed by a comparison of the predicted means of the Housing factor. Data are individual values and mean ± SEM.

## Histology

A variable amount of bone was seen in the defects, as were remnants of the graft material in group CPC. In group AB, newly formed lamellar bone could not be distinguished from the grafted bone. Three rabbits out of four in group AB-Floor, presented only bone filling the defect with cancellous bone, including both woven and lamellar appearances as well as adult

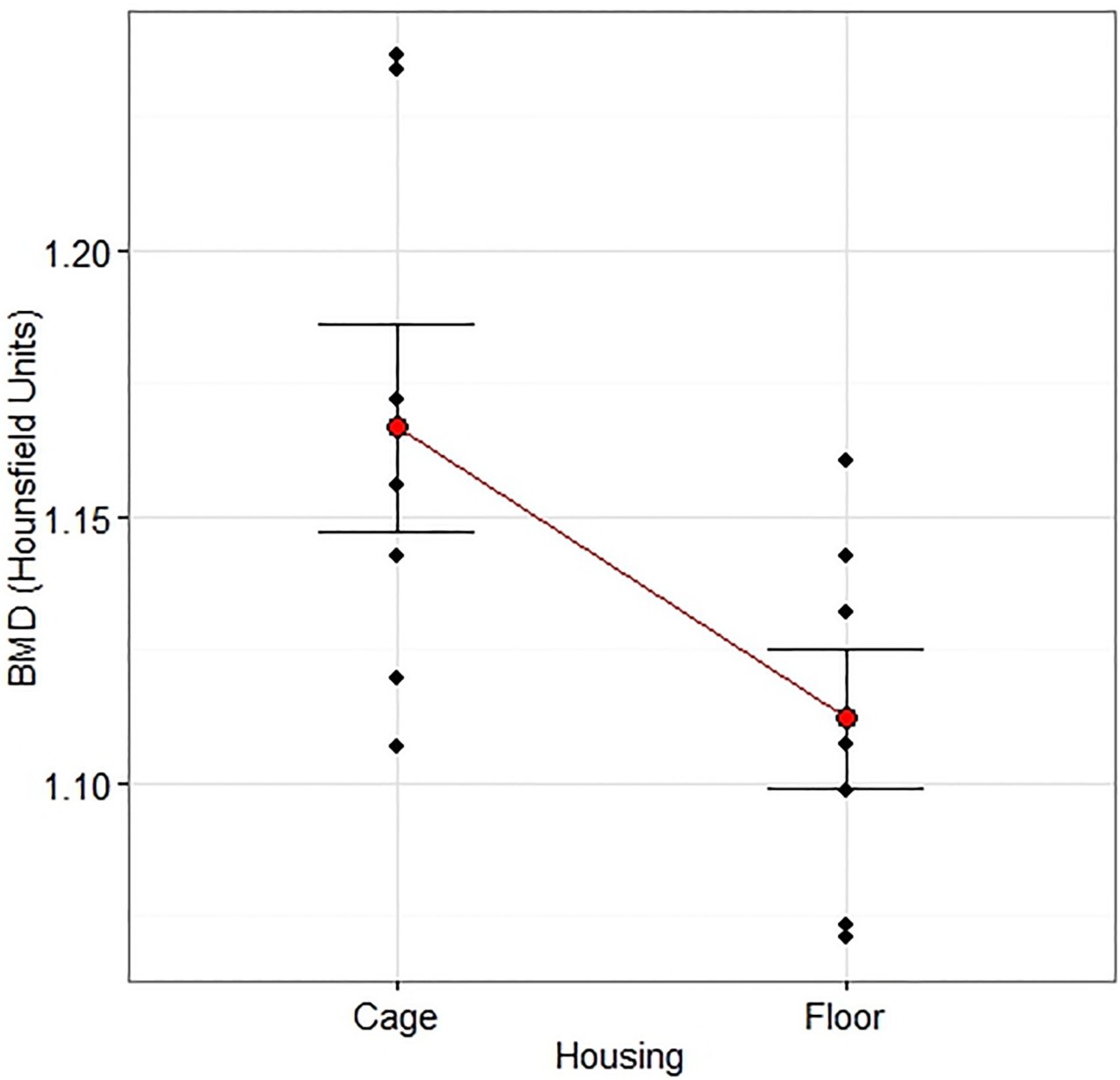

**Fig 12. Density measured by μ-CT.** The effect of Housing on the defect density measured by micro-CT 12 weeks after creation of a critical radius defect treated with a calcium phosphate compound or augmented bone. The measurement was performed along the entire defect length (20 mm) and included the ulnar bone. There was an overall effect of Housing (p = 0.044, F = 5.34) that did not interact with Treatment (p = 0.85, F = 0.04). The density was lower in floor housed rabbits (n = 7) than cage housed (n = 7), CIs [1.08, 1.15], [1.13, 1.21], p = 0.043 by two-way ANOVA with Housing and Material as treatment factors followed by a comparison of the predicted means of the Housing factor. Data are individual values and mean ± SEM.

type of fatty bone marrow inter-trabecular (Fig 13a and 13b). In group AB-Cage however, three out of four rabbits also showed areas of fibrocartilage in the woven and lamellar bone, resulting in an area without complete bone healing (Fig 13c and 13d). There were however no significant differences between individual scores for any category except for category g (bony fusion between the ulna and the radius adjacent to the defect), which was larger in group Floor (p = 0.045). Median sum of scores did not differ (see S1 Dataset).

**Table 3. Volume and density measured by μ-CT in the central area of the radius defect, 12 w after creation (Op 10), and of the contralateral uninjured side (Intact 10 and 20).** Measurements were made in the central part of the defect (10 mm), or the whole defect (20 mm). The measurements of the defect filling include the adjacent ulna and any surrounding new bone. There was no difference between group Floor and Cage in volume or density in the defect area or the intact area (two-way-ANOVA with factor Housing displayed).

| | | Cage (n = 7) | | Floor (n = 7) | | | |
|---|---|---|---|---|---|---|---|
| | | Mean | SEM | Mean | SEM | P-value | F-value |
| Op 10 | Volume (mm³) | 198 | 17 | 230 | 10 | 0.09 | 0.02 |
| | Density (Hounsfield units) | 1.17 | 0.02 | 1.11 | 0.01 | 0.09 | 3.5 |
| Intact 10 | Volume (mm3) | 184 | 3 | 185 | 4 | 0.9 | 0.02 |
| | Density (Hounsfiled units) | 1.30 | 0.04 | 1.31 | 0.03 | 0.63 | 0.63 |
| Intact 20 | Volume (mm3) | 367 | 10 | 371 | 10 | 0.78 | 0.08 |
| | Density (Hounsfiled units) | 1.30 | 0.01 | 1.32 | 0.01 | 0.42 | 0.72 |

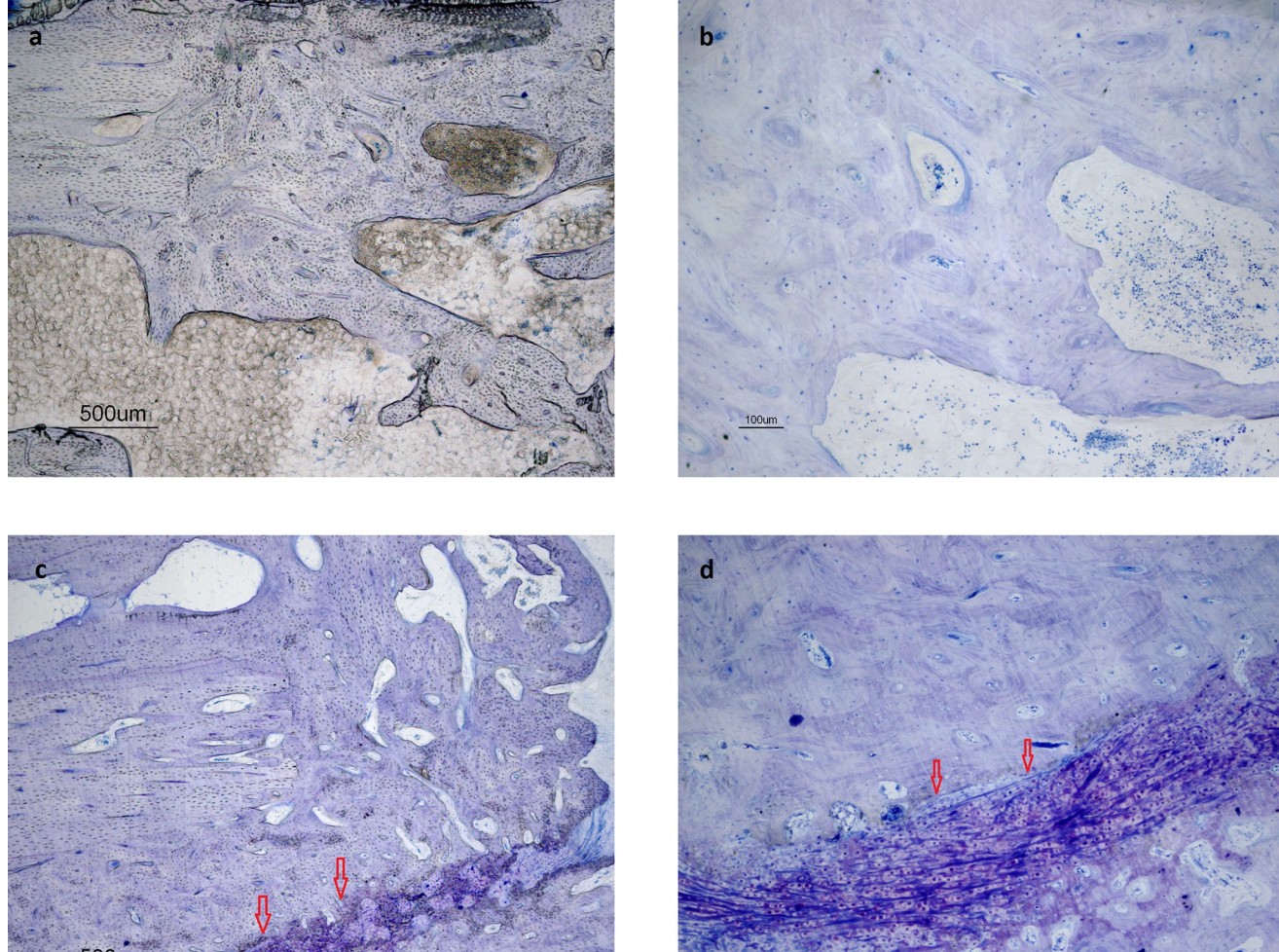

**Fig 13.** a-b. Rabbit in group AB-Floor: areas of complete reorganisation of trabecular bone. This represents a score 4 for category c (cancellous bone) and a score 3 for category e (filling of the defect). c-d. Rabbit in group AB-Cage: area of incomplete healing with fibrocartilage (arrows). This represents a score 2 (filling of the defect) for category e.

(a)

(b)

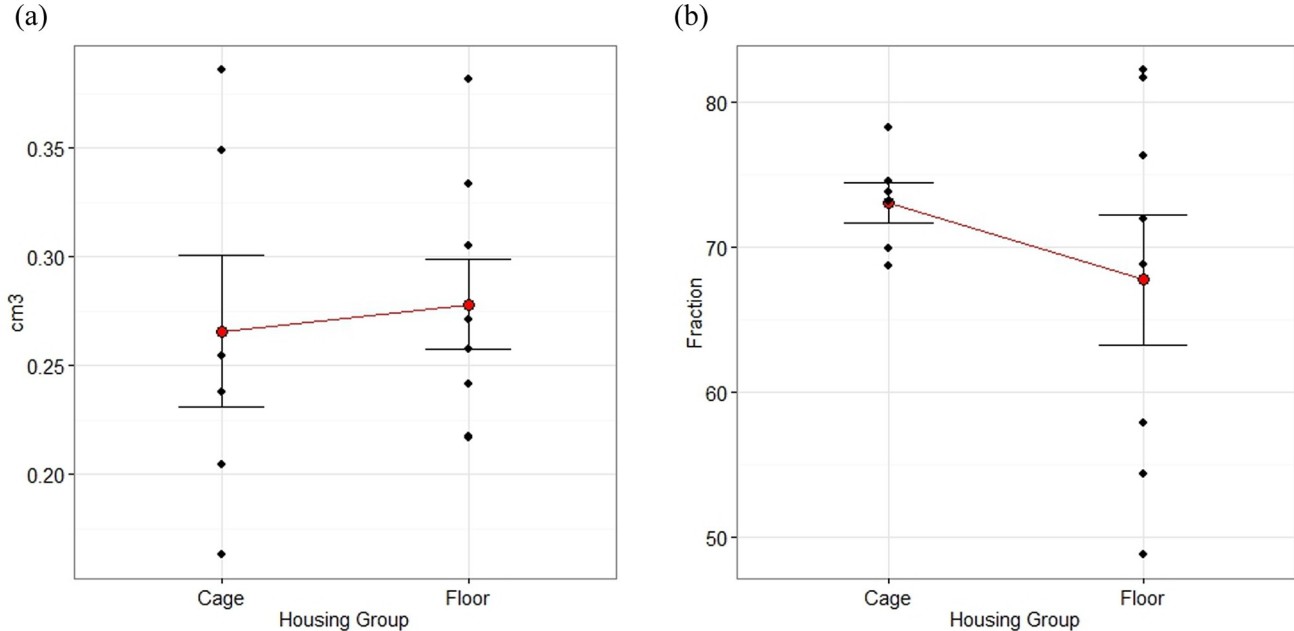

**Fig 14.** a-b. Mean with SEM plots of total area and fraction of area filled with bone and material as measured by histomorphometry, 12 weeks after creation of a radius defect filled with calcium phosphate compound block or autologous bone. The area of measurement was outlined by the defect ends, the ulnar bone and the graft material + new bone. Rabbits were housed in pairs in floor pens of 3 m$^2$ (n = 7) or singly in cages of 0.43 m$^2$ (n = 7). NS by Mann-Whitney test.

## Histomorphometry

There were no differences in the total area between groups Floor and Cage, nor in the area filled by graft material and bone (Fig 14a and 14b).

## Discussion

The study was conducted to evaluate the influence of housing conditions on the experimental treatment in a critical radius defect in rabbits. The rabbit radius defect was chosen as model because it is recommended by international standards for pre-clinical evaluation in critical segmental bone defects [20]. The anatomy of the rabbit ulna makes external fixation material superfluous, and full load can be placed on the leg directly after surgery.

The rabbits were purchased at the age of 8–9 months from a breeding with cage housing (single or in pairs). At the experimental facility, the rabbits were either single housed in standard cages (group Cage), or pair housed in 7 times larger floor pens (group Floor). Rabbits in group Floor showed an approximate 30% higher activity of CS than those in group Cage, which reflects an increased physical activity. CS activity in skeletal muscle has been shown to increase by 28% with physical activity in rats housed in activity cages [21] and by 14–24% in rabbits after 4–6 w of housing in pens with obstacles to jump over [22].

A higher physical activity probably explains the 18% increase in defect volume, i.e. new bone, of the defects in floor housed rabbits in the present study, as well as the higher degree of bony fusion with reorganisation of trabecular bone, between the ulna and the radius adjacent to the defect. Further, histology showed that there were regions of incomplete bone healing in rabbits in group AB-Cage, in contrast to rabbits in group AB-Floor.

Physical activity is known to influence the proliferation and differentiation of mesenchymal progenitor cells into chondrocytes and osteoblasts in the callus [23]. Apart from stimulating healing, the risk for osteopenia with subsequent osteoporosis is reduced with physical activity [24]. Especially peak activity like running and jumping, which cause peak strain on leg bones has been shown to contribute to cortical bone thickness in rats [25] and increased bone mass and density in humans [26]. Further, loading of bone increases the blood flow, which is important for the delivery of growth factors, which in turn regulate bone healing [27]. The defect mineral density was 4% lower in group Floor, which can be explained by the fact that active bone remodelling includes a high bone turnover that may result in less bone mineral.

For the translational value of animal studies, experiments should be performed under physiological conditions, and the human clinical condition closely mimicked. The loading of bone can influence the outcome in fracture repair [28] and is usually encouraged in humans with fractures. The reporting of rabbit orthopaedic studies often lacks detailed information on housing conditions, despite recommendations on how to report animal studies [29]. In the experience of the authors, rabbits in most medical research are housed singly in standard cages. Cage housing has been reported to cause inactivity atrophies in the proximal femur [30], although space requirements have increased since these studies were performed in the 1980s. A larger cage size has been shown to increase cortical thickness, bone diameter and weight, but not strength in rabbits [8, 9].

Pair and group housing of rabbits increases physical activity However, increased activity does not automatically indicate increased welfare, since it can be related to aggressive behaviour causing escape behaviour. Under laboratory conditions, crowding can lead to competition, increase in stress hormone levels and a higher infection pressure [31, 32]. Adult males fight rigorously, and females may even prefer a solitary pen, regardless of rank [33]. In the current study, two rabbit pairs needed re-grouping, due to fighting. The death of one rabbit before study begin (gut dysbacteriosis) may have been related to stress caused by co-housing. However, single housing in cages is also known to cause stress, as well as boredom [34]. Cage housed rabbits lie and sit more often than rabbits kept in larger floor pens and floor housed rabbits show more rearing, digging, stretching, hopping, and running behaviours. Additional studies are needed to evaluate whether the best alternative for long-term orthopaedic studies would be single housing in floor pens, with contact to rabbits in adjacent housing. By providing different types of enrichment within the housing area (cover, hay, gnawing sticks, hiding areas, platforms), locomotion is stimulated, and boredom reduced.

A limitation of the present study is the low number of rabbits. Only one rabbit was lost due to reasons related to the experimental model; the fractured ulna in the largest rabbit. Nevertheless, it was possible to identify differences in bone healing that were dependent on the housing conditions. Another weakness of the study was that new bone could not be differentiated from the grafted bone. The identification of new bone can be achieved by injection of fluorescent markers like tetracycline or calcein in vivo [35].

In summary, the presented study indicates that pair housing in larger floor pens leads to an increase in physical activity and increased fusion with formation of mature lamellar bone in a created bone defect in rabbits, when compared with single housing in cages.

## Conclusion

Bone healing in rabbit models should be studied under conditions that are as physiological as possible. Restricting the locomotion by housing in standard cages may negatively influence the outcome of rabbit orthopaedic studies.

## Supporting information

**S1 Dataset.**
(XLSX)

## Acknowledgments

Thanks to Birgitta Essén and Karin Söderlund who performed the muscle CS activity measurements.

## Author Contributions

**Conceptualization:** Patricia Hedenqvist, Torbjörn Mellgren, Marianne Jensen-Waern, Andreas Thor.

**Data curation:** Patricia Hedenqvist, Caroline Öhman-Mägi, Petra Hammarström Johansson, Stina Ekman, Cecilia Ley.

**Formal analysis:** Patricia Hedenqvist, Amela Trbakovic, Torbjörn Mellgren, Petra Hammarström Johansson, Stina Ekman, Cecilia Ley.

**Funding acquisition:** Patricia Hedenqvist, Marianne Jensen-Waern.

**Investigation:** Patricia Hedenqvist, Amela Trbakovic, Torbjörn Mellgren, Petra Hammarström Johansson, Elin Manell, Stina Ekman, Cecilia Ley, Marianne Jensen-Waern, Andreas Thor.

**Methodology:** Patricia Hedenqvist, Amela Trbakovic, Torbjörn Mellgren, Caroline Öhman-Mägi, Petra Hammarström Johansson, Stina Ekman, Marianne Jensen-Waern, Andreas Thor.

**Project administration:** Patricia Hedenqvist.

**Resources:** Patricia Hedenqvist, Caroline Öhman-Mägi, Andreas Thor.

**Supervision:** Patricia Hedenqvist, Caroline Öhman-Mägi, Andreas Thor.

**Visualization:** Patricia Hedenqvist, Torbjörn Mellgren, Stina Ekman, Cecilia Ley.

**Writing – original draft:** Patricia Hedenqvist, Amela Trbakovic, Torbjörn Mellgren, Petra Hammarström Johansson, Stina Ekman, Cecilia Ley.

**Writing – review & editing:** Patricia Hedenqvist, Amela Trbakovic, Torbjörn Mellgren, Caroline Öhman-Mägi, Petra Hammarström Johansson, Elin Manell, Stina Ekman, Cecilia Ley, Marianne Jensen-Waern, Andreas Thor.

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
