## [Decision Letter · Decision Letter 0]

10 Mar 2020

PONE-D-20-00117

The effect of housing space on bone healing in a critical radius defect in New Zealand White rabbits

PLOS ONE

Dear Dr. Hedenqvist,

Thank you for submitting your manuscript to PLOS ONE. After careful consideration, we feel that it has merit but does not fully meet PLOS ONE’s publication criteria as it currently stands. Therefore, we invite you to submit a revised version of the manuscript that addresses the points raised by  the reviewer and editorial comments during the review process.

We would appreciate receiving your revised manuscript by Apr 24 2020 11:59PM. To enhance the reproducibility of your results, we recommend that if applicable you deposit your laboratory protocols in protocols.io, where a protocol can be assigned its own identifier (DOI) such that it can be cited independently in the future. For instructions see: http://journals.plos.org/plosone/s/submission-guidelines#loc-laboratory-protocols

We look forward to receiving your revised manuscript.

Kind regards,

Dr. Sakamuri V. Reddy

Academic Editor

PLOS ONE

Additional Editor Comments (if provided):

The authors studied the effect of housing space on bone healing in a critical bone defect model using rabbits. I suggest to rephrase the title “….housing environment….” . There have been reports on load bearing effects on intact bones in rabbits, but the effect of housing conditions is unknown. However, bone healing in a critical defect animal models are studied using the scaffold delivery of various factors. Therefore, I suggest the authors begin the Abstract with an introduction sentence giving a rationale with housing space/other conditions/increased mobility etc variability in studies. Abstract sections conclude that larger housing space increased physical activity and promoted bone formation. I suggest removing the repetitive part of the conclusion “suggesting that housing condition is an important factor in bone healing studies.” Introduction-section should be detailed of studies in other animal models also. Results-(line 259) They should delete the unpublished report/manuscript in preparation that “The effects of the material are presented elsewhere (Trbakovic/Mellgren et al, in manuscript)” and may include in the discussion on possible variability in effects observed as data not shown. There is no need to include citrate synthase assay method as supplemental figure as they have given citations (14-15) in Methods section.

Journal Requirements:

Reviewers' comments:

Reviewer's Responses to Questions

**Comments to the Author**

1. Is the manuscript technically sound, and do the data support the conclusions?

Reviewer #1: Yes

2. Has the statistical analysis been performed appropriately and rigorously? 

Reviewer #1: Yes

3. Have the authors made all data underlying the findings in their manuscript fully available?

Reviewer #1: Yes

4. Is the manuscript presented in an intelligible fashion and written in standard English?

Reviewer #1: Yes

5. Review Comments to the Author

Reviewer #1: The authors describe a critical size defect model in rabbits investigating the influence of the cage size on bone regeneration.

Abstract:

The abstract is written well and contains all relevant information.

Introduction:

The introduction is written well and leads to the topic.

The fact that physical activity increases BMD and decreases the risk of osteoporosis is well known and does from my point of view not really need extra animal studies.

At the end of the introduction and hypothesis was formed.

Page 3, line 43, please use numbers for quotation all the time

Methods:

In general the study protocol is not so clear to me. Why using 2 different scaffolds in very small groups to compare the cage size. It seems the study was initially planned for another purpose, also comparing the scaffolds.

Loosing 30% of the rabbits due to complications is bad. You should really think about your methods.

Methods are described well and with sufficient details. Surgical procedure is also described like performed on a high level.

How did you differ between bone of the rabbit, CPC and bone graft by µCT?

You got a high bone volume with a low bone density for the animal in the larger cages and in contrast a low bone Volume with a high bone mineral density for the animals with the smaller cages?

Can this be a matter of thresholding? I seems like you are counting more callus with low bone density in the large cage group than in the small cage group.

Due to the low number of surviving animals and the high standard deviation of Bone volume and histological data I feel your study is underpowered to answer the effect of the cage size on bone regeneration and bone mineral density.

line 67: only female rabbits - influence of oestrogen (could be mentioned in discussion)

line 98: the implant was based on one CT. Did it fit accurately on all the rabbits? Did any dislocations occur (besides fractures)?

line 98: please add the measurements of the implants

line 141: please add a photo or sketch of the surgical procedure (especially the fixation)

line 162: limping scored by one observer- Is this significant? Would it be better if it was observed by more than one person ( 2 - 3 blinded observers)?

line 164: Was this the same observer as before? If not, how can we compare the inter-individual differences?

line 201: subjective scoring: as said in line (204) blinding was not possible, is there an informative value?

line 261: two or three weeks acclimatization (compared to line 72)?

Results:

Results as well as the lost animals are described well.

You should indicate that there is significance in figure 6.

line 284: which score did you use

Discussion:

The discussion is written well. I would wish to get a few more sentences regarding the 6/20 lost rabbits. I think this is way higher than the average of other studies.

line 403: single housing means stress - could there be higher cortisol levels, therefore resulting in lower bone formation?

line 411: limitations were clearly stated

missing: Why is the bone density lower in the floor group (results line 300)? Bone density and volume of the defect size increase, when animals are active. (https://doi.org/10.1016/j.injury.2016.05.037)

Conclusion:

The conclusion is appropriate.

6. PLOS authors have the option to publish the peer review history of their article (what does this mean?). If published, this will include your full peer review and any attached files.

Reviewer #1: No

---

## [Author Response · Author response to Decision Letter 0]

21 Apr 2020

Response to reviewers

PONE-D-20-00117 

The effect of housing space on bone healing in a critical radius defect in New Zealand White rabbits

Dear Dr. Reddy,

Thank you for the invitation to submit a revised version of the manuscript. I want to thank you and the reviewer for the valuable comments and suggestions, which improve the manuscript substantially. The points raised by the reviewer and editorial comments are addressed below. 

Editorial comments

The authors studied the effect of housing space on bone healing in a critical bone defect model using rabbits. I suggest to rephrase the title “….housing environment….” . Changed as suggested. Line 1. 

There have been reports on load bearing effects on intact bones in rabbits, but the effect of housing conditions is unknown. However, bone healing in a critical defect animal models are studied using the scaffold delivery of various factors. Therefore, I suggest the authors begin the Abstract with an introduction sentence giving a rationale with housing space/other conditions/increased mobility etc variability in studies. Changed as suggested. Lines 21-22

Abstract sections conclude that larger housing space increased physical activity and promoted bone formation. I suggest removing the repetitive part of the conclusion “suggesting that housing condition is an important factor in bone healing studies.” Changed as suggested. Line 39.

Introduction-section should be detailed of studies in other animal models also. Has been added. Lines 42-44 and 60-61.

Results-(line 259) They should delete the unpublished report/manuscript in preparation that “The effects of the material are presented elsewhere (Trbakovic/Mellgren et al, in manuscript)” and may include in the discussion on possible variability in effects observed as data not shown. Has been deleted. Line 273. 

There is no need to include citrate synthase assay method as supplemental figure as they have given citations (14-15) in Methods section. Removed. Line 544.

Please ensure that your manuscript meets PLOS ONE's style requirements, including those for file naming. The PLOS ONE style templates can be found at http://www.plosone.org/attachments/PLOSOne_formatting_sample_main_body.pdf and http://www.plosone.org/attachments/PLOSOne_formatting_sample_title_authors_affiliations.pdf

2. We note that you have included the phrase “data not shown” in your manuscript. Unfortunately, this does not meet our data sharing requirements. PLOS does not permit references to inaccessible data. We require that authors provide all relevant data within the paper, Supporting Information files, or in an acceptable, public repository. Please add a citation to support this phrase or upload the data that corresponds with these findings to a stable repository (such as Figshare or Dryad) and provide and URLs, DOIs, or accession numbers that may be used to access these data. Or, if the data are not a core part of the research being presented in your study, we ask that you remove the phrase that refers to these data. Data was added to Table 3 and the supporting raw data to the uploaded data file. Line 330.

Reviewer #1: 

The authors describe a critical size defect model in rabbits investigating the influence of the cage size on bone regeneration.

Abstract:

The abstract is written well and contains all relevant information.

Introduction:

The introduction is written well and leads to the topic.

The fact that physical activity increases BMD and decreases the risk of osteoporosis is well known and does from my point of view not really need extra animal studies. 

At the end of the introduction and hypothesis was formed.

Page 3, line 43, please use numbers for quotation all the time The mistake has been corrected. Line 46.

Methods:

In general the study protocol is not so clear to me. Why using 2 different scaffolds in very small groups to compare the cage size. It seems the study was initially planned for another purpose, also comparing the scaffolds. It is correct that the study was designed for comparing the scaffolds and simultaneously the effect of housing conditions. We hypothesized that the scaffolds would not interfere with physical activity and with the two-way analysis an interference would be detected. The aim was to get as much out of one study as possible. 

Loosing 30% of the rabbits due to complications is bad. You should really think about your methods. We admit that the mortality was extreme, and we haven’t lost any rabbit (0/62) in our previous orthopaedic and anaesthesia studies (Hedenqvist et al 2013, 2014a, 2014b, 2016). A stated, there were multiple reasons for the losses, which we think are important to report. To prevent complications from clotting, we intend to use coated catheters in the future. 

Methods are described well and with sufficient details. Surgical procedure is also described like performed on a high level.

How did you differ between bone of the rabbit, CPC and bone graft by µCT? The CPC could easily be distinguished from bone because of a higher density, but new bone could not be distinguished from the grafted bone, as commented in lines 440-1. 

You got a high bone volume with a low bone density for the animal in the larger cages and in contrast a low bone Volume with a high bone mineral density for the animals with the smaller cages? Correct, according to the micro-CT evaluation.

Can this be a matter of thresholding? No. A standard reference material was scanned each day in order to accurately compare the samples between measurements and across sample groups (line 199-201). I seems like you are counting more callus with low bone density in the large cage group than in the small cage group. No. No consideration for maturity of bone was made. All the material above the threshold values was measured in each case. 

Due to the low number of surviving animals and the high standard deviation of Bone volume and histological data I feel your study is underpowered to answer the effect of the cage size on bone regeneration and bone mineral density. While it is often desirable to have more animals, we calculated 20 rabbits to be sufficient for a two-way design. Despite the unfortunate losses, there were statistical significances which we feel are important to report. 

line 67: only female rabbits - influence of oestrogen (could be mentioned in discussion). The focus of the study was not to address sex differences, and not possible this time. The rabbits were not neutered, so there was no lack of estrogen. 

line 98: the implant was based on one CT. Did it fit accurately on all the rabbits? Did any dislocations occur (besides fractures)? The surgical prefabricated guide allowed for surgical removal of an exact amount of bone repeated in every case, well corresponding to the prefabricated implants. Other than fractures, part of the CPC implants were slightly dislocated. Added to text, line 313-4. Fig 8 added. Line 320-1

line 98: please add the measurements of the implants Added, line 141.

line 141: please add a photo or sketch of the surgical procedure (especially the fixation). Two photos have been added (Fig 2a & b). Lines 149-152.

line 162: limping scored by one observer- Is this significant? Would it be better if it was observed by more than one person ( 2 - 3 blinded observers)? This is correct, we have deleted this part and only generally described the effects of surgery on limping. Lines 171-176 and 296-302.

line 164: Was this the same observer as before? If not, how can we compare the inter-individual differences? Part has been deleted. Lines 300-302.

line 201: subjective scoring: as said in line (204) blinding was not possible, is there an informative value? You are right, we have deleted this part. Lines 350-355.

line 261: two or three weeks acclimatization (compared to line 72)? Corrected. 

Results:

Results as well as the lost animals are described well.

You should indicate that there is significance in figure 6. Corrected. Now Fig 7. 

line 284: which score did you use Removed. Lines 350-355.

Discussion:

The discussion is written well. I would wish to get a few more sentences regarding the 6/20 lost rabbits. I think this is way higher than the average of other studies. We agree that the loss was very high, but we think that an elaboration does not add to the aim of the study, especially since there were many different causes for the losses. 

line 403: single housing means stress - could there be higher cortisol levels, therefore resulting in lower bone formation? It is indeed possible that stress may have had an additional impact, however the data shows a significant effect of space & pair-housing on the activity of citrate synthase (CS), which is directly correlated to physical activity. 

line 411: limitations were clearly stated

missing: Why is the bone density lower in the floor group (results line 300)? Bone density and volume of the defect size increase, when animals are active. (https://doi.org/10.1016/j.injury.2016.05.037) In an early stage of bone healing with a large proportion of woven bone, the density can be lower. The density is expected to increase at a later stage, when remodeling is complete and access callus has been absorbed. Comment added to discussion, lines 408-10.

Conclusion:

The conclusion is appropriate.

---

## [Decision Letter · Decision Letter 1]

7 May 2020

The effect of housing environment on bone healing in a critical radius defect in New Zealand White rabbits

PONE-D-20-00117R1

Dear Dr. Hedenqvist,

We are pleased to inform you that your manuscript has been judged scientifically suitable for publication and will be formally accepted for publication once it complies with all outstanding technical requirements.

With kind regards,

Dr. Sakamuri V. Reddy

Academic Editor

PLOS ONE

Additional Editor Comments (optional):

Reviewers' comments:

Reviewer's Responses to Questions

**Comments to the Author**

1. If the authors have adequately addressed your comments raised in a previous round of review and you feel that this manuscript is now acceptable for publication, you may indicate that here to bypass the “Comments to the Author” section, enter your conflict of interest statement in the “Confidential to Editor” section, and submit your "Accept" recommendation.

Reviewer #1: All comments have been addressed

2. Is the manuscript technically sound, and do the data support the conclusions?

Reviewer #1: Yes

3. Has the statistical analysis been performed appropriately and rigorously? 

Reviewer #1: Yes

4. Have the authors made all data underlying the findings in their manuscript fully available?

Reviewer #1: Yes

5. Is the manuscript presented in an intelligible fashion and written in standard English?

Reviewer #1: Yes

6. Review Comments to the Author

Reviewer #1: Dear authors,

Thank you for revising the manuscript.

I think it has improved substantially, congratulations!

7. PLOS authors have the option to publish the peer review history of their article (what does this mean?). If published, this will include your full peer review and any attached files.

Reviewer #1: No

---

## [Editor Report · Acceptance letter]

11 May 2020

PONE-D-20-00117R1 

The effect of housing environment on bone healing in a critical radius defect in New Zealand White rabbits 

Dear Dr. Hedenqvist:

I am pleased to inform you that your manuscript has been deemed suitable for publication in PLOS ONE. Congratulations! Your manuscript is now with our production department. 

With kind regards,

on behalf of

Dr. Sakamuri V. Reddy 

Academic Editor

PLOS ONE